# The Relationship between Anxiety and Depression Levels and General Health Status before and 12 Months after SUI Treatment in Postmenopausal Women from the Lower Silesian Population

**DOI:** 10.3390/ijerph19095156

**Published:** 2022-04-24

**Authors:** Maciej Zalewski, Gabriela Kołodyńska, Felicja Fink-Lwow, Anna Mucha, Waldemar Andrzejewski

**Affiliations:** 1Department of Gynaecology and Obstetrics, Faculty of Health Sciences, Medical University of Wrocław, 50-367 Wrocław, Poland; zalewskim@interia.pl; 2Department of Gynaecology, Independent Public Health Care Center of the Ministry of the Interior and Administration in Wroclaw, 50-233 Wrocław, Poland; 3Department of Physiotherapy, Wroclaw University of Health and Sport Sciences, 51-612 Wrocław, Poland; felicitas1@wp.pl (F.F.-L.); waldemar.andrzejewski@awf.wroc.pl (W.A.); 4Department of Genetics, Wrocław University of Environmental and Life Sciences, 50-375 Wrocław, Poland; anna.mucha@upwr.edu.pl; 5Faculty of Health Sciences, University of Opole, 45-040 Opole, Poland

**Keywords:** depression, anxiety, stress urinary incontinence

## Abstract

Menopause is often the cut-off point from which most cases of stress urinary incontinence (SUI) in women begin. This dysfunction affects not only the physical experience of the patient, but is also related to the psychological aspects, leading to a reduced quality of life. Despite the large number of patients with SUI and the frequent use of surgical treatment for this condition, there are few scientific reports evaluating the effectiveness of the procedure in terms of reducing depressive symptoms or improving overall health. The aim of this study was to evaluate the relationship between anxiety and depression and general health status before and 12 months after surgical treatment for SUI in postmenopausal women. Seventy-five patients qualified for the study, but due to the long study duration, both sets of questionnaires were eventually obtained from 60 postmenopausal patients. All patients that qualified for the study had a trans obturator tape (TOT) procedure. All patients enrolled in the project were given the Hospital Anxiety and Depression Scale (hAdS) and King’s Health Questionnaire (KHQ). After 12 months of surgery with midurethral slings, symptoms of depression were present in only a small number of subjects, 11.7%, and anxiety was present in 13.3% of the entire group. The study confirms that patients with a general poor health condition may suffer from depression or anxiety, and therefore may also need psychological treatment. Patients with SUI should therefore receive therapeutic care from a multidisciplinary team, in which therapeutic activities are divided between doctors, nurses, physiotherapists and psychologists. As a result of the treatment, after 12 months, we confirmed a significant improvement in patients with depression and anxiety disorders.

## 1. Introduction

The World Health Organization (WHO) and the International Continence Society (ICS) define urinary incontinence (UI) as an involuntary leakage of urine from the bladder that is not only a hygienic and social problem, but also a psychological one, most often affecting women [1,2]. The number of people with this condition is not fully known, partially caused by a low reporting rate due to the private nature of urinary dysfunction. Available data on the prevalence of UI in the population are conflicting. According to some authors, 56–69% of women experience one or more incontinence episodes at least once a year [3,4]. The causes are varied, although the most common type of incontinence is SUI, estimated at 50 to 88% of all UI cases [5]. Other factors that increase the risk of this condition include: age, pregnancy, natural childbirth, obesity, diabetes and menopause. Menopause is often the cut-off point from which most cases of SUI in women begin [6,7,8]. This dysfunction affects not only the physical experience of the patient, but is also related to psychological aspects, leading to a reduced quality of life [9,10].

Quality of life is an extremely important aspect of the existence of every human being. Delkey and Rurke define it as a sense of well-being, satisfaction or dissatisfaction with life, i.e., happiness or unhappiness [11]. Nowadays, more and more attention is paid to this issue because it is widely known that it is an important tool for measuring the health of the population. It is often the basis of planning health education and creating preventive programs and projects aimed toward health planning [12]. The issue of quality of life is closely related to the definition of health itself, which, according to the World Health Organization (WHO), means not only the absence of disease or disability, but also the state of physical, social and mental well-being [13]. Currently, doctors are assessing patients’ health increasingly often, in addition to biometric indicators, taking into account a subjective assessment of a patient’s own health. Moreover, a significant correlation was found between the mental and physical health of patients [14].

Depression and anxiety are common conditions affecting between 2 and 16.5% of the population [15]. As with UI, the exact number of patients with this problem is unknown, primarily due to the failure to properly diagnose patients. However, this is known to be a civilization problem with increasing trends [16]. At the same time, it is often closely associated with the presence of other chronic conditions. The relationship between anxiety and depression and lower urinary tract symptoms (LUTS) has been confirmed by numerous researchers over the years. Particular attention is paid to the correlation with stress urinary incontinence [17]. A study by Coyne et al. found that 49.7% of women with SUI have anxiety, while depression is present in 34.9% of women with this condition [18]. Numerous studies also confirm that patients with SUI more often suffer from depression and anxiety compared to women without stress urinary incontinence symptoms [19,20,21,22]. There are also scientific reports indicating that using incontinence treatments reduces the incidence of depression and anxiety [23,24,25,26].

The treatment of urinary incontinence, regardless of the etiology, should begin with a conservative treatment based primarily on physiotherapy and lifestyle modifications [27]. Surgical treatment is recommended in cases of significant disease progression or when physiotherapy is not effective. There are more than a dozen primary types of surgery performed to treat SUI. The selection of an appropriate technique in each case should be individual, dictated by factors related to the patient, as well as the operator’s experience or equipment availability. One of the most commonly used techniques is the trans obturator tape procedure (TOT), which involves inserting a synthetic tape under the middle section of the urethra, between the obturator foramina [28,29].

Despite the large number of patients with SUI and the frequent use of TOT in the surgical treatment of this condition, there are few scientific reports evaluating the effectiveness of the procedure in terms of reducing depressive symptoms or improving overall health. It seems that apart from the assessment of the reduction in disease severity as a result of the performed procedure, the assessment of the psychological aspects should also be the center of interest for every clinician or physiotherapist starting treatment for patients with SUI. The analysis of the effects of surgery on the psychological status of patients, especially for a long-term assessment, also seems relevant.

The aim of this study was to evaluate the relationship between anxiety and depression (by HADS), and general health status (by KHQ), before and 12 months after surgical treatment for SUI in postmenopausal women.

## 2. Materials and Methods

### 2.1. Design and Data Collection

Seventy-five patients were qualified for the study, but due to the long study duration, both sets of questionnaires were eventually obtained from 60 postmenopausal patients. The age of the examined patients ranged from 65 to 87 years (mean 68.83 years, median—69 years, SD—6.26). The women who qualified for the study had a BMI in the range of 17.63–38.86 kg/m^2^ (mean 27.07 kg/m^2^, median 26.74 kg/m^2^, SD—4.69), body height 1.49–1.72 m (mean—1.61 m, median—1.62 m, SD—0.05), while body weight 48–100 kg (mean—69.94 kg, median—69.5 kg, SD—11, 65).

Each patient who qualified for the study was interviewed about their history of gynecological and urological diseases. In the interview, questions about obstetric history, including the number and type of births, were asked. The vast majority of patients gave birth by force of nature—91.67%. The number of deliveries in the study group ranged from 1 to 4 (median = 2). The patients also provided information on professional and sports activity. Sixty-five percent of respondents declared that they regularly engage in physical activity, mostly because 95% of women are currently retired or on a pension. The interview also asked about the presence of comorbidities that may affect urinary incontinence, where 81.67% of women indicated that they had comorbidities. The most common comorbidities were hypertension and diabetes.

All patients who qualified for the study had a TOT procedure at the gynecology department in the Hospital of the Ministry of Internal Affairs and Administration in Wroclaw. Additionally, all patients gave their informed written consent to take part in the study prior to entering the project. All participants completed the Hospital Anxiety and Depression Scale (HADS), and the King’s Health Questionnaire (KHQ) twice: once upon admission to the ward, and 12 months after surgery. The patients qualified for the study on the basis of the inclusion and exclusion criteria.

The inclusion criteria were:Stage III stress urinary incontinence confirmed by ultrasound examination and an interview;Not taking hormone replacement therapy (HRT) before or after the surgical procedure;Informed written consent of the patient to take part in the project.

The exclusion criteria were:Women with overactive bladders (OAB) or mixed urinary incontinence (MUI);Women with urinary tract fistulas;Women with congenital or acquired defects of the urethra or the bladder;Women with urinary tract infections;Women taking medication that contributes to an overactive bladder.

### 2.2. Measures

All the patients enrolled in the project were given the Hospital Anxiety and Depression Scale and King’s Health Questionnaire. The questionnaires were completed upon admission to the department, before the TOT procedure, and 12 months after the procedure at the follow-up examination. The number of responses to particular questions varied, as some patients marked two or none of the responses that they felt best described their quality of life at that time. 

The Hospital Anxiety and Depression Scale (HADS) is a validated tool for detecting anxiety and depression in non-psychiatric patients. The HADS scale contains 3 independent subscales: depression (D), anxiety (A) and aggression (R), consisting of a total of 16 questions (each can be scored from 0 to 3 points). In total, a subject can score a maximum of 48 points (this is the sum of the maximum number of points from each subscale). In a summary assessment, an individual’s conditions can be described as: 0–16 points—no disturbance; 17–22 points—borderline states; 23–48 points—occurrence of the disorder. The HADS has been validated in the Polish language [30,31].King’s Health Questionnaire (KHQ)—The structure of the KHQ questionnaire contains 26 questions that are divided into 3 parts. The first part (Part I) consists of 2 questions and is used to assess women’s general health status and to analyze the effect of SUI symptoms on mental status. The second part (Part II) contains 16 questions to assess the effect of urinary symptoms on five areas of life. There are seven sub-domains addressing: RE (Role Emotional), RP (Role Physical), SL (Social Limitations), PR (Personal Relationships), E (Emotions), S/E (Sleep/Energy), and SS (Severity of Symptoms). The third part (Part III) uses 6 questions to assess the degree of difficulty resulting from the need to protect against incontinence symptoms. These difficulties are associated with, e.g., the need to wear pads or frequent changes of underwear. A larger number of points obtained as a result of the questionnaire means a worse QoL for the patient. A maximum of 8 points may be scored in Part I, 64 points in PART II, and 24 points in Part III. The KHQ has been validated in the Polish language [32,33].

## 3. Statistical Analysis

Statistical analysis was performed using R Project software version 4.1.3 created by RStudio, Vienna, Austria. The statistical significance of the surgery’s effect on the patient’s general health and the level of depression was tested. The basic descriptive statistics of the domain scores were determined using the *pastecs* package [34]. The statistical significance of the surgery’s effect on the domain scores was verified with a Wilcoxon signed-rank test for dependent samples. The statistical significance of depression’s effect on the domain scores was verified with a Wilcoxon signed-rank test for independent samples. The statistical significance of the surgery and depression effect on the domain scores was verified using a Kruskal–Wallis nonparametric analysis of variance. This method was used because of the non-compliance of the trait distribution considered with normal distribution. After the Kruskal–Wallis test, multiple comparison tests were performed using the *agricolae* package [35]. The statistical significance of the effect of the analyzed groups on the frequency of individual responses to the questions in the questionnaires considered was verified using Fisher’s exact test. The relationship between the domain scores was investigated using Spearman’s rank correlation with the *psych* package [36]. The correlation in which 100% (1 − *α/k*) confidence intervals did not overlap was considered statistically significantly different between groups [37]. The significance level for all statistical tests was *p* < 0.05.

## 4. Results

In our study, 75 patients met the inclusion criteria, of which 60 completed the KHQ (Part I, II and III) and HADS questionnaires correctly before and 12 months after surgery, and thus were included in our study. The characteristics of the study group are shown in Table 1.

A detailed analysis of the data in relation to the individual parts of the KHQ before surgery showed that, in relation to the KHQ, in Part I (concerning general health and the effect of SUI symptoms on mental status), 35% of patients obtained the maximum unfavorable score (8 points). On the other hand, unfavorable results for Part II (related to the effect of urinary dysfunction on other areas of functioning) were obtained by 38.3%, and for Part III (related to difficulties in eliminating the somatic effects of dysfunction) by 60% of women. This result allowed us to conclude that a significant proportion of patients presented an unfavorable health status in all assessed domains before surgery. In contrast, a similar analysis for the HADS-A showed no disorders in 53.3%, borderline cases in 23.3%, and abnormal cases in 23.3% of subjects. In the HADS-D assessment, the absence of disorders was demonstrated by 66.7%, borderline cases by 20% and abnormal cases by 13.3% of patients.

Table 2 and Figure 1 show the differences in the scores of the analyzed HADS domains (i.e., A—anxiety, D—depression, R—aggression) before and 12 months after the treatment.

There was a statistically significant reduction in feelings of anxiety (A) and depression (D) (*p* < 0.05) 12 months after the surgery. However, no statistically significant changes were observed with respect to HADS-R (R—aggression).

A detailed analysis of HADS-A and HADS-D subscales scores before and 12 months after the surgery only confirmed a statistically significant improvement, presented by a higher number of women experiencing no anxiety vs. anxiety despite surgery (*p* = 0.0484) for HADS-A (Table 3 and Figure 2).

Statistically significant reductions in KHQ values were observed for Parts I, II, and III, comparing data before and 12 months after treatment, respectively (Table 4 and Figure 3), indicating a beneficial effect in all studied domains.

The analysis of three domains of the KHQ—general health (PART I), associations of urinary dysfunction with other areas of life (PART II), and difficulties arising from incontinence in everyday life (PART III) in terms of the severity of anxiety symptoms—described by the HADS-A subscale before surgery is presented in Table 5, and at 12 months after the surgery in Table 6. There was a statistically significant association between the self-assessment of general health according to the KHQ (Part I), and the severity of anxiety based on the HADS-A subscale (*p* < 0.01131) before the surgery. No significant associations were shown for the other two domains. 

In addition, a similar analysis performed 12 months after the treatment confirmed statistically significant association for the second domain, associated with RE, RP, SL, PR, E, S/E and SS. There was a statistically significant association between the first and the second domains of the KHQ questionnaire and the severity of anxiety based on the HADS-A subscale, *p* < 0.001024 and *p* < 0.0195, respectively (Table 6).

Table 7 shows the results of all three KHQ questionnaire domains with respect to the severity of the disorder, based on the HADS-D subscale in the assessment carried out after surgery. There was a statistically significant association between all three parts of the KHQ questionnaire, and the HADS-D subscale, for each part, at *p* < 0.0108, *p* < 0.004216, *p* < 0.002061, respectively.

To sum up the analysis, we evaluated the most important components affecting the results.

Table 8 shows the eigenvalues, cumulative percentage of explained variability and loadings of the first four principal components of PCA (principal component analysis) for the KHQ subscales and HADS subscales. Due to the eigenvalues (>1), the first two principal components, which explain 73.88% of the variability in the study population, were selected for further analysis. It can also be seen that none of the analyzed variables showed a high correlation with the third or fourth principal components. The first principal component (PC1) is most strongly and negatively correlated with individual parts of the KHQ (except for GHP, i.e., general health perception). The second principal component (PC2) is strongly and negatively correlated with the three HADS subscales (i.e., A—anxiety, D—depression, R—aggression).

## 5. Discussion

Menopause, in most cases, is a natural biological process, during which the production of estrogen decreases and eventually stops. Initially, it is associated with the occurrence of irregular menstrual cycles and, after some time, their final stoppage. This period is associated with characteristic symptoms, such as hot flushes and night sweats. The following are common symptoms: fatigue and apathy, lowered mood, irritability, nervousness, the impaired ability of concentration and memory, sleep disorders, vaginal dryness, intercourse-related difficulties, impaired libido, and stress urinary incontinence [38].

There are studies that have demonstrated that stress urinary incontinence occurs in up to 50% of postmenopausal patients [39]. SUI negatively affects many aspects of patients’ lives. The symptoms of incontinence not only affect physical experience, but also have a very strong effect on mental health. SUI is an embarrassing problem that is not mentioned by most women to their loved ones or medical staff. Because of incontinence, patients are very often forced to change their lifestyle. They often withdraw from social life or even avoid social contact. The accompanying ailments also negatively affect their intimate relationships. They are often left alone with their problem, which adversely affects their quality of life, causing the appearance of depressive symptoms and states of anxiety.

The appearance or worsening of depressive symptoms in the postmenopausal period is associated with fluctuations in estrogen levels. It is also affected by changes in many areas of family, work and social life. It is very common for women during this period to experience a change in their perception of their body image and sense of femininity. Menopause has been found to be particularly conducive to exacerbation or the recurrence of previously diagnosed depression, but also that which occurs over the course of bipolar affective disorder. Depressive symptoms in this period are characterized by a specific clinical picture. They very often require a different management than the treatment of depression in women of other age groups. Depression occurring in the postmenopausal period adversely affects quality of life and functioning in women [40].

Many cross-sectional studies were conducted on large patient groups, showing that 100% of women with UI suffer from depression. These include a study by Kaur et al., in which all 100 women with UI enrolled in the project were found to have depression. Among those, 48% suffered from severe depression, and 45% from moderately severe depression [41]. Similar results were presented by Zorn et. al., where 115 patients reported significant depression [42]. Moghaddas et al. showed a 52% prevalence of depression in UI women of 50–64 years of age [43]. In contrast, Morrison et al. observed severe depression in only 11.6% of women with UI. Nevertheless, they concluded that it was an extremely serious and common condition in patients with UI and required prompt diagnosis and treatment to avoid serious consequences [44].

Our study using HADS-D also showed the presence of depressive symptoms in 33.3% of the subjects with SUI (borderline case, 20%, vs. abnormal case, 13.3%). After surgery with midurethral slings, which resulted in a reduction in SUI symptoms in all patients, depressive symptoms were present only in a small number of subjects, i.e., 11.7% (HADS-D) of the entire group. This allowed us to conclude that there was a reduction in the number of patients with depressive symptoms 12 months after surgery in our study group of postmenopausal women.

The only study that assessed the depression level in patients after midurethral slings surgery was that by Siff et al. For their study, they enrolled 526 female patients with SUI, of whom 79 (15%) initially showed symptoms of severe depression. The authors showed that the severity of incontinence symptoms was higher in these women than in non-depressed women. In addition, depressed patients had a reduced quality of life and sexual function. Twelve months after surgery, 83% of women previously diagnosed with depression presented no symptoms. Their results showed no association of depression with symptom severity, quality of life or sexual function [45].

In our study, in addition to assessing the prevalence of depression in patients before and 12 months after surgery, we also determined the effect of SUI symptoms on the overall quality of life assessed by the KHQ questionnaire. Previous studies showed that women with stress urinary incontinence experienced a significant reduction in quality of life. Saboia et al. included 556 female patients with different types of incontinence in their study. These authors used different questionnaires in their evaluation, yet the KHQ was still one of them. According to researchers, the mixed urinary incontinence (MUI) form has the greatest effect on decreasing general health perception. In addition, patients with SUI also experienced a significant reduction in quality of life across all assessed scores [46].

The validity of the King’s Health Questionnaire (KHQ) in patients before and after surgical treatment for urinary incontinence was assessed by Luz et al. For their study, they enrolled 204 patients who underwent incontinence surgery with transobturator MUS between 2004 and 2013. Assessments were performed 6, 12, and 24 months after surgery. The results showed that the KHQ questionnaire was a valuable assessment tool after UI surgery and determined clinically relevant threshold scores for defining subjective outcomes [47].

Based on our results from the KHQ questionnaire, it can be concluded that most of the patients presented an unfavorable health status in all assessed domains at the time of being found eligible for surgery. Twelve months after surgery, the results were significantly improved in all domains, i.e., general health, as well as selected spheres of life and everyday functioning. We also demonstrated a significant association between all three domains of the KHQ questionnaire (I, II, III), and the degree of depressive symptoms according to HADS-D.

In the light of the reported data on depression in women with UI, the International Continence Society recently suggested that clinical trials of patients with UI should also include an assessment of quality of life, rather than just determining the severity of symptoms or type of incontinence. Currently, medical personnel are becoming increasingly aware of the presence of depression in patients with SUI, which may affect the course of the therapy itself. According to this, patients’ mental health should be assessed both before and after surgery, as it should not be considered a priori that eliminating incontinence symptoms would automatically improve patients’ quality of life and mental state. The results of the present study allow us to conclude that the majority of postmenopausal women with stress urinary incontinence from Lower Silesian Voivodeship presented an unfavorable health status in all assessed domains of the KHQ questionnaire before the surgery. A similar situation was observed in the assessment of depression, as the majority of patients (33.3%) showed a significant percentage of the disorder before treatment.

This study showed a significant association between anxiety (A) and depression (D), especially with the general health status (GHP) assessed in Part I of the KHQ questionnaire before the treatment. This relationship raises concerns and requires further research. The study confirms that patients with a generally poor health condition may suffer from depression or anxiety, and therefore may also need psychological treatment. Patients with SUI should therefore receive therapeutic care from a multidisciplinary team, in which therapeutic activities are divided between doctors, nurses, physiotherapists and psychologists. As a result of the treatment, after 12 months, we confirmed a significant improvement in 43.3% (HADS-A) and 17% (HADS-D) of patients with depression and anxiety disorders.

## 6. Conclusions

The majority of postmenopausal women with stress urinary incontinence from Lower Silesian Voivodeship presented an unfavorable health status in all assessed domains of the KHQ questionnaire before surgery.A similar situation was observed in the assessment of anxiety and depression, as 33.3% of patients showed significant levels of depression before the treatment.This study showed a significant association between anxiety and depression, specifically general health status, as assessed by the KHQ questionnaire before the treatment.After 12 months of surgery with midurethral slings, symptoms of depression were present in only a small number of subjects, i.e., 11.7% (HADS-D), and anxiety was present in 13.3% (HADS-A) of the entire group.

## 7. Limitations

Our paper has certain limitations. The group was too small, and we only studied women over the age of 65, which requires future follow-up on a larger number of women, including premenopausal women. On the other hand, the group was homogeneous in terms of professional activity (post-working age), as well as cultural and social conditions (female residents of Lower Silesian Voivodeship who used a specific health care facility in Wrocław). We only studied the TOT treatment, and thus our data cannot be extended to different slings or to other type of treatments. We used comprehensive measures, the King’s Health Questionnaire and the Hospital Anxiety and Depression Scale. We acknowledge that choosing other validated questionnaires might have provided additional depression and QoL information.

## Figures and Tables

**Figure 1 ijerph-19-05156-f001:**
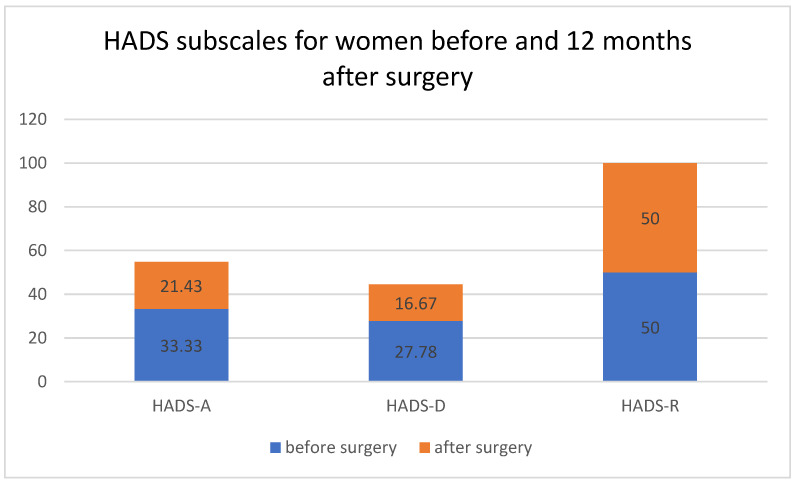
HADS subscales for women before and 12 months after surgery. Medians differed between HADS-A, HADS-D and HADS-R (A—anxiety, D—depression, R—aggression, Hospital Anxiety and Depression Scale) before and 12 months after surgery.

**Figure 2 ijerph-19-05156-f002:**
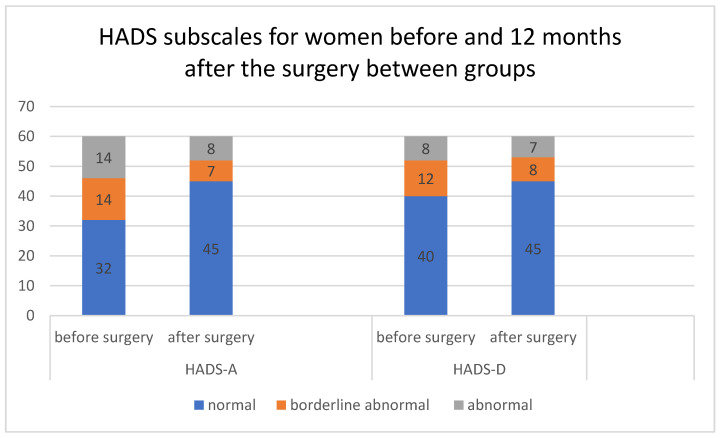
HADS subscales for women before and 12 months after the surgery between groups. Number of individuals in HADS-A and HADS-D (A—anxiety and D—depression, Hospital Anxiety and Depression Scale) before and 12 months after surgery.

**Figure 3 ijerph-19-05156-f003:**
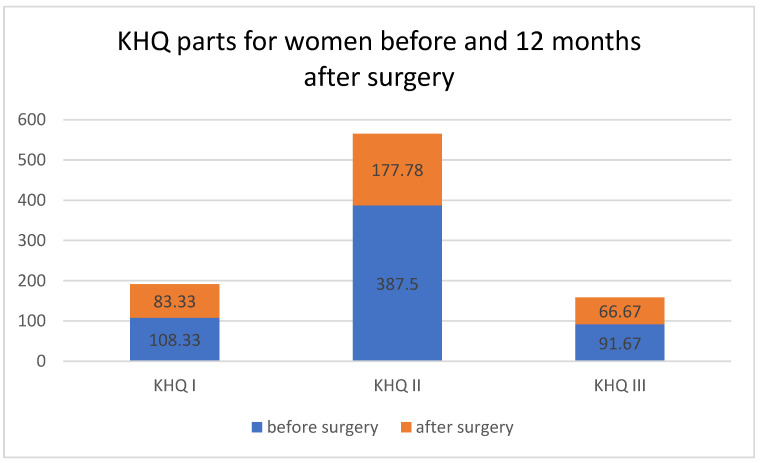
KHQ parts for women before and 12 months after surgery. Medians differing in Part I, Part II and Part III of the King’s Health Questionnaire (KHQ) before and 12 months after surgery.

**Table 1 ijerph-19-05156-t001:** Characteristics of the studied group before surgery.

Patients (*n*)	60
Age (years) median (min–max)	69 (65–87)
Weight (kg) median (min–max)	69.5 (48–100)
Body height (m) median (min–max)	1.62 (1.49–1.72)
BMI (kg/m^2^) median (min–max)	26.74 (17.63–38.86)
Postmenopausal (%)	100
KHQ Part I median (min–max)	108.33 (25.00–200.00)
KHQ Part II median (min–max)	387.50 (0.00–700.00)
KHQ Part III median (min–max)	91.67 (50.00–150.00)
HADS-A score	33.33 (0–100)
HADS-D score	27.78 (0–100)
HADS-R score	50 (0–100)

*n* = number of individuals, min = minimum, max = maximum, BMI = body mass index, KHQ = King’s Health Questionnaire, HADS = Hospital Anxiety and Depression Scale, D—depression, A—anxiety, R—aggression.

**Table 2 ijerph-19-05156-t002:** HADS subscales for women before and 12 months after surgery.

Statistics	Subscale					
	HADS-A		HADS-D		HADS-R	
	Before Surgery	After Surgery	Before Surgery	After Surgery	Before Surgery	After Surgery
** *n* **	60	60	60	60	60	60
**Minimum**	0	0	0	0	0	0
**Maximum**	100	100	100	100	100	100
**Median**	33.33 ^a^	21.43 ^b^	27.78^a^	16.67 ^b^	50	50
**Mean**	35.48	26.98	33.33	25.83	49.72	50
**SD**	21.32	23.16	24.03	25.36	30.30	28.95
**VC (%)**	60.10	85.82	72.10	98.18	60.94	57.90
**Effect size**	0.538 (large)		0.482 (moderate)		0.051 (small)	

*n* = number of individuals, SD = standard deviation, VC = variation coefficient (presented as a percentage). Differing medians that were statistically significant between groups are marked by different letters (Wilcoxon test, *p*-value < 0.05).

**Table 3 ijerph-19-05156-t003:** HADS subscales for women before and 12 months after the surgery between groups were marked by different letters.

Subscale	Surgery	Normal	Borderline Abnormal (Borderline Case)	Abnormal (Case)	*p*-Value
**HADS-A**	Before	32	14	14	0.0484
	After	45	7	8	
**HADS-D**	Before	40	12	8	0.5514
	After	45	8	7	

Data are presented as subgroups by size. Given *p*-values are for the Fisher’s exact test. A—anxiety, D—depression.

**Table 4 ijerph-19-05156-t004:** KHQ parts for women before and 12 months after surgery.

Statistics	PART					
	I		II		III	
	Before Surgery	After Surgery	Before Surgery	After Surgery	Before Surgery	After Surgery
** *n* **	60	60	60	60	60	60
**Minimum**	25.00	0.00	0.00	0.00	50.00	50.00
**Maximum**	200.00	200.00	700.00	611.11	150.00	150.00
**Median**	108.33	83.33	387.50	177.78	91.67	66.67
**Mean**	110.28	77.50	336.99	205.23	89.86	78.47
**SD**	47.60	48.74	190.44	196.88	27.92	28.96
**VC (%)**	43.16	62.90	56.51	95.93	31.07	36.91
***p*-value** **Effect size**	0.00004564 0.532 (large)		0.00003793 0.526 (large)		0.00319 0.407 (moderate)	

*n* = number of individuals, SD = standard deviation, VC = variation coefficient (presented as a percentage). *p*-value and effect size refer to Wilcoxon test for dependent variables.

**Table 5 ijerph-19-05156-t005:** KHQ—PART I, II, III sub-scale scores for women before the surgery, with different disorder stages defined based on the HADS-A subscale.

KHQ Subscale	HADS-A Disorder Level	Statistics							
		Minimum	Maximum	Median	Mean	SD	VC (%)	*p*-Value	Effect Size
**Part I**	Normal (*n* = 32)	25.00	200.00	91.67 ^a^	96.35	48.36	50.19	0.01131	0.122 (moderate)
	Borderline abnormal (*n* = 14)	58.33	150.00	108.33 ^ab^	107.74	28.77	26.70		
	Abnormal (*n* = 14)	50.00	200.00	162.50 ^b^	144.64	46.06	31.85		
**Part II**	Normal (*n* = 32)	0.00	700.00	295.83	293.06	190.38	64.93	0.0722	0.0571 (small)
	Borderline abnormal (*n* = 14)	16.67	602.78	376.39	357.34	178.34	49.91		
	Abnormal (*n* = 14)	0.00	616.67	463.89	417.06	185.10	44.38		
**Part III**	Normal (*n* = 32)	50.00	150.00	83.33	83.07	24.82	29.88	0.07648	0.0551 (small)
	Borderline abnormal (*n* = 14)	50.00	150.00	95.83	95.24	34.55	36.27		
	Abnormal (*n* = 14)	50.00	133.33	104.17	100.00	24.89	24.89		

*n* = number of individuals, SD = standard deviation, VC = variation coefficient (presented as a percentage). The groups of demonstrated disorder level that were statistically significant and different in terms of analyzed KHQ subscale are marked by different letters. Given *p*-value is for Kruskal—Wallis test.

**Table 6 ijerph-19-05156-t006:** KHQ—Parts I, II, III sub-scale scores for women 12 months after the surgery, with different disorder stages defined based on the HADS-A subscale.

KHQ Subscale	HADS-A Disorder Level	Statistics							
		Minimum	Maximum	Median	Mean	SD	VC (%)	*p*-Value	Effect Size
**Part I**	Normal (*n* = 45)	0.00	150.00	58.33 ^a^	63.33	44.40	70.10	0.001024	0.206 (large)
	Borderline abnormal (*n* = 7)	83.33	175.00	116.67 ^b^	119.05	33.58	28.21		
	Abnormal (*n* = 8)	83.33	200.00	112.50 ^b^	120.83	38.83	32.14		
**Part II**	Normal (*n* = 45)	0.00	611.11	66.67 ^a^	159.44	174.45	109.41	0.0195	0.103 (moderate)
	Borderline abnormal (*n* = 7)	122.22	602.78	369.44 ^b^	363.49	167.17	45.99		
	Abnormal (*n* = 8)	0.00	611.11	350.00 ^ab^	324.31	238.19	73.44		
**Part III**	Normal (*n* = 45)	50.00	150.00	66.67	74.07	27.88	37.64	0.07677	0.055 (small)
	Borderline abnormal (*n* = 7)	58.33	141.67	91.67	90.48	25.65	28.35		
	abnormal (*n* = 8)	50.00	150.00	91.67	92.71	33.46	36.10		

*n* = number of individuals, SD = standard deviation, VC = variation coefficient (presented as a percentage). The groups of demonstrated disorder level that were statistically significant and different in terms of the analyzed KHQ subscales are marked by different letters. Given *p*-value is for Kruskal—Wallis test.

**Table 7 ijerph-19-05156-t007:** KHQ—Parts I, II, III sub-scale scores for women 12 months after the surgery, with different disorder stages defined based on the HADS-D subscale.

KHQ Subscale	HADS-D Disorder Level	Statistics							
		Minimum	Maximum	Median	Mean	SD	VC (%)	*p*-Value	Effect Size
**Part I**	Normal (*n* = 45)	0.00	150.00	75.00 ^a^	65.56	45.35	69.18	0.0108	0.124 (moderate)
	Borderline abnormal (*n* = 8)	50.00	175.00	112.50 ^b^	113.54	38.04	33.50		
	Abnormal (*n* = 7)	58.33	200.00	100.00 ^ab^	113.10	48.08	42.51		
**Part II**	Normal (*n* = 45)	0.00	602.78	66.67 ^a^	150.68	167.06	110.87	0.004216	0.157 (large)
	Borderline abnormal (*n* = 8)	0.00	611.11	413.89 ^b^	384.38	189.88	49.40		
	Abnormal (*n* = 7)	0.00	611.11	322.22 ^ab^	351.19	210.84	60.03		
**Part III**	Normal (*n* = 45)	50.00	141.67	58.33 ^a^	70.74	25.23	35.66	0.002061	0.182 (large)
	Borderline abnormal (*n* = 8)	50.00	150.00	100.00 ^b^	100.00	28.87	28.87		
	Abnormal (*n* = 7)	66.67	150.00	108.33 ^b^	103.57	28.41	27.43		

*n* = number of individuals, SD = standard deviation, VC = variation coefficient (presented as a percentage). The groups of demonstrated disorder level that were statistically significant and different in terms of analyzed KHQ subscales are marked by different letters. Given *p*-value is for Kruskal—Wallis test.

**Table 8 ijerph-19-05156-t008:** Eigenvalues, the percentage of cumulative explained variance and loadings of the first four principal components of PCA of KHQ subscales and HADS subscales.

	PC1	PC2	PC3	PC4
**Eigenvalue**	8.90	2.18	0.80	0.69
**Cumulative variance percentage**	59.34	14.54	5.35	4.60
**Loadings**				
GHP	−0.50	−0.57	0.50	−0.36
II	−0.85	0.20	0.19	0.36
**KHQ-Part I**	−0.88	−0.14	0.39	0.09
RL	−0.89	0.20	0.10	0.14
PL	−0.84	0.25	−0.06	0.22
SL	−0.85	0.08	−0.25	−0.24
PR	−0.71	0.22	−0.35	−0.13
E	−0.91	0.10	−0.05	−0.14
SE	−0.78	−0.01	0.00	−0.22
SM	−0.86	0.22	0.05	0.05
**KHQ-Part II**	−0.97	0.18	−0.10	−0.05
**KHQ-Part III**	−0.84	0.12	−0.02	−0.10
HADS-A	−0.49	−0.75	−0.10	0.22
HADS-D	−0.49	−0.72	−0.12	0.14
HADS-R	−0.32	−0.67	−0.36	0.02

GHP = general health perception, II = incontinence impact, RL = role limitations, PL = physical limitations, SL = social limitations, PR = personal relationships, E = emotions, SE = sleep/energy, SM = severity measures, PCA = principal components analysis.

## Data Availability

The datasets used and/or analyzed during the current study are available from the corresponding author upon reasonable request.

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
