# Peer review of "The Relationship between Anxiety and Depression Levels and General Health Status before and 12 Months after SUI Treatment in Postmenopausal Women from the Lower Silesian Population"

_ijerph, 2022, doi:10.3390/ijerph19095156_

Round 1
Reviewer 1 Report
Thank you very much for the interesting text carried out on Polish patients.
The text is suitable for publication after introducing minor corrections in order to improve the quality of the text.
- The authors on lines 48, 110 - write about the quality of life. I think that the very aspect of assessing the quality of life in relation to health should be completed at least in a short paragraph. There are many authors who study the phenomenon of living standards in relation to health, e.g.ZióÅ‚kowska-Weiss, K., Assessment of the Selected Health Factors by Polonia in the Greater Toronto Area in the Context of Quality and Standard of Living, International Journal of Environmental Research and Public Health. 2021; 18(3):1296. 1-20, https://doi.org/10.3390/ijerph18031296, Power, M. J.; Quinn, K.; Schmidt, S.; WHOQOL-OLD Group. Development of the WHOQOL-OLD module. Qual. Life Res. 2005, 14, 2197–2214., Alonso, J.; Ferrer, M.; Gandek, B.; Ware, J. E.; Aaronson, N. K.; Mosconi, P. Health-related quality of life associated with chronic conditions in eight countries: Results from the International Quality of Life Assessment (IQOLA) Project. Qual. Life Res. 2004, 13, 283–298.
Please expand this thread
- Line 84 - the characteristics of the examined persons should be extensive. In fact, the authors write about this in limitation, writing that the studies were limited, that the age of the subjects was over 65, but this information should be included in section 2.1. If the authors have the characteristics of the respondents, these data should be included and described in the description (for example, information about the age of the oldest patient)
Author Response
Dear reviewer, thank you very much for your help and your time. Your comments significantly improved the quality of our manuscript. Following your suggestions, we have implemented all the changes you have suggested in the manuscript
As suggested, we added a quality-of-life paragraph that significantly enriched the content of the article. In addition, we also added the characteristics of our patients.
Thank you.
Kind regards,
Authors

Reviewer 2 Report
- The abbreviation used in the text is confused. All abbreviations must be defined upon first mention in the body of the manuscript.
- Table content is not intuitive and clear, most be improved. It would be better to consider using figures to present the results.
- The results part must be improved.
- What does TVT mean in the line of 353? According to the methods part, all patients qualified for the study had a TOT procedure. Please confirm it.
Author Response
Dear reviewer, thank you very much for your help and your time. Your comments significantly improved the quality of our manuscript. Following your suggestions, we have implemented all the changes you have suggested in the manuscript
We have improved the abbreviations throughout the text.
In addition, in the results section, we have added figures that actually better visualize the reported results.
Of course, in the line of 353 we made a mistake for which we sincerely apologize and thank you for pointing it out. We confirm that all patients underwent the TOT procedure.
Thank you.
Kind regards,
Authors

Round 2
Reviewer 2 Report
1 The abbreviations used in the text are still not standardized, please revise it carefully.
2 The type of figure 1 and 3 is not appropriate, please choose a more appropriate chart type to present the results.
3 In the introduction part, the author used a non-English lauguage.
Author Response
Dear reviewer,
Thank you very much for your help and time for reviewing our manuscript.
1. As suggested by the reviewer, we have corrected the description of the abbreviations used in the manuscript.
2. As suggested, we corrected Figure 1, which for some unknown reason contained the same data as in Figure 3. Additionally, we added descriptions with abbreviations under all tables and figures. We would like to thank you especially for this remark, because if the reviewer had not indicated it, we would have published incorrect data.
3. We submitted the text for a linguistic revision, which significantly improved the quality of the manuscript.
Thank you once again.
Kind regards,
Authors
